



# Water level variation at a beaver pond significantly impacts net CO₂ uptake of a continental bog

Hongxing He[1], Tim Moore[1], Elyn R Humphreys[2], Peter M Lafleur[3], and Nigel T Roulet[1]

[1]Department of Geography, McGill University, Montreal, Quebec H3A OB9, Canada
[2]Geography and Environmental Studies, Carleton University, Ottawa, ON, Canada
[3]School of Environment, Trent University, Peterborough, ON, Canada

*Correspondence to*: Hongxing He (hongxing-he@hotmail.com)

**Abstract.** The carbon (C) dynamics of northern peatlands are sensitive to hydrological changes owing to ecohydrological feedback. We quantified and evaluated the impact of water level variations in a beaver pond (BP)

on the CO₂ flux dynamics of an adjacent, raised *Sphagnum* – shrub-dominated bog in southern Canada. We applied the CoupModel to the Mer Bleue bog, where the hydrological, energy and CO₂ fluxes have been measured continuously for over 20 years. The lateral flow from the bog to the BP was estimated by the hydraulic gradient between the peatland and the BP's water level and the vertical profile of peat hydraulic conductivity. The model outputs were compared with the measured hydrological components, CO₂ flux and energy flux data (1998-2019).

CoupModel was able to reproduce the measured data well. The simulation shows that variation in the BP water level (naturally occurring or due to management) influenced the bog net ecosystem exchange of CO₂ (NEE). Over 1998-2004, the BP water level was 0.75 to 1.0 m lower than during 2017-2019. Simulated net CO₂ uptake was 55 g C m⁻² yr⁻¹ lower during 1998-2004 compared to 2017-2019 when there was no BP disturbance, which was similar to the differences in measured NEE between those periods. Peatland annual NEE was well correlated with

water table depth within the bog, and NEE also shows a linear relation with the water level at the BP, with a slope of -120 g CO₂-C m⁻² yr⁻¹ m⁻¹. The current modelling predicts the bog may switch from CO₂ sink to source when the BP water levels drop lower than ~ 1.7 m below the peat surface at the eddy covariance tower, 250 m from the BP. This study highlights the importance of natural and human disturbances to adjacent water bodies in regulating net CO₂ uptake function of northern peatlands.

## 25  1 Introduction

Northern peatlands cover only 3% of the global land area but contain ~ 30% of the soil carbon (C) (Yu et al., 2014). Pristine peatlands, characterized by waterlogged situations, accumulate CO₂ from the atmosphere as partially decayed vegetation, owing to the anoxic and cool conditions constraining the decomposition of organic matter (Gorham, 1991). Multi-year field measurements indicate that northern peatlands are currently sinks for

CO₂ with a net uptake rate between 0 to 80 g C m⁻² yr⁻¹ (~ 0.1 to 0.5 Pg C yr⁻¹ globally) (Roulet et al., 2007; Nilsson et al., 2008; Dinsmore et al., 2010; Helfter et al., 2015). However, drying of peatlands by natural or anthropogenic disturbance could potentially weaken the sink strength or release the historically stored peat back into the atmosphere (Wu and Roulet, 2014; Qiu et al., 2020).

The net ecosystem exchange (NEE) of peatlands depends on the balance between CO₂ uptake through

photosynthesis and losses through respiration, both of which are closely regulated by hydrology (Hilbert et al., 2000; Silvola et al., 1996; Wallén et al., 1988). In general, hydraulic conductivity is high in the upper aerobic



layer of the peat (i.e., acrotelm), but decreases as the plant material becomes more decomposed in the lower layer that is permanently saturated (i.e., catotelm) (Ingram, 1978; Clymo, 1992). When water storage decreases, the water table falls, lateral flow and mass inputs decrease, so the peat surface drops relative to the water table (Belyea and Baird, 2006). Conversely, when water storage increases, the water table rises, increasing the difference between net primary production and decomposition, so the mass input and peatland height increase. This feedback between the C dynamics and water storage is key to peatland self-regulation (Eppinga et al., 2009; Frolking et al., 2010). However, external disturbance, for example changing hydrological boundary conditions, can potentially push the peatland outside its limits for self-regulation (Harris et al., 2020).

In North America, boreal peatlands are often associated with streams and beaver ponds (BP) (Rosell et al., 2005; Rebertus, 1986; Loisel et al., 2017). Beavers (*Castor canadensis* in North America and *C. fiber* in Eurasia) have been residing in such environments for thousands of years (Naiman et al., 1988). Identified as a keystone species and ecosystem engineers, beavers can modify the hydrological boundary surrounding a peatland by constructing dams in channels and maintaining higher water levels at the peatland margins. Beaver activities are shown to affect hydrological processes such as flow path alteration (Naiman et al., 1988), water storage-runoff relations (Tardif et al., 2009) and peatland water table dynamics (Karran et al., 2018). Consequently, the integrity of beaver dams and their ability to regulate water levels and peatland hydrology (Westbrook et al., 2020) may impact the C dynamics of the adjacent peatlands. Despite the extent of beavers in peatland regions and their profound impact on hydrology, the effect of altering the hydrological boundary conditions as BP water levels on the NEE of adjacent peatlands has not, to our knowledge, been quantified.

Mer Bleue is a raised-shrub bog located in southern Canada that has been studied extensively for C gas exchanges and biogeochemistry (e.g., Lafleur et al. (2003), Roulet et al. (2007), Moore and Bubier (2019)). In the early years of work at Mer Bleue, the beaver numbers and beaver ponds surrounding the bog were managed by the landowner (the National Capital Commission, NCC), which lowered the water level at the BP. However, after 2005 the NCC abandoned their beaver management program resulting in BP water level rises, providing an opportunity to examine whether changing boundaries conditions alter the $CO_2$ dynamics at the ecosystem scale.

This study aims to evaluate and quantify the effect of changes in peatland water storage due to water level variations (i.e., changing lateral boundary conditions) in the BP adjacent to Mer Bleue bog. To link the varying boundary conditions and the associated $CO_2$ dynamics, a process-based model, CoupModel, was used. We chose CoupModel (Jansson, 2012; He et al., 2021) because it includes detailed soil physics, hydrology, photosynthesis, and ecosystem respiration that can simulate the association of the peatland C dynamics with vertical and lateral hydrology flow (He et al., 2016). Furthermore, past studies have shown the model can successfully simulate the C dynamics and the energy and water balance of managed European treeless peatlands (Metzger et al., 2015), but as it has not been applied to continental bogs. This study aims to evaluate CoupModel for continental bog conditions using a long (~ 21 year) continuous dataset of hydrology and flux observations at Mer Bleue. The model outputs were first compared with the measured data, and the validated model was then used to analyze the effect of BP water levels on the peatland's $CO_2$ fluxes.

## 2 Site and Methods

### 2.1 Site description

75    Mer Bleue is a raised, ombrotrophic bog located 10 km east of Ottawa, Canada (45.41°N, 75.48°W). The climate
of the region is cool temperate continental, with a mean annual temperature of $6.2 \pm 0.8$ °C and annual precipitation
of $955 \pm 159$ mm (235 mm fell as snow) during the studied period 1998-2019. These values were similar to the
30 year (1971-2000) mean of 6.0 °C and 943 mm in the region, respectively. The peatland is slightly domed with
peat depths varying from 5 to 6 m near the center decreasing to < 0.3 m at the margin with a narrow band of

beaver ponds surrounding the bog (Fig. 1). The part of Mer Bleue in the footprint area of the eddy covariance (EC)
tower has an average peat thickness of ~ 3.4 m. The studied area is locally flat with hummock – hollow
microtopographic differences of ~ 0.25 m (Wilson, 2012). Hummocks cover ~ 70% of the surface area. There is
a slight slope (0.08%) from the center of the peatland to the margins. The average elevation of the peat surface is
69.7 AMSL.

Vegetation on hummocks is dominated by evergreen (*Chamaedaphne calyculata, Rhododendron groenlandicum,
Kalmia angustifolia*), and sparse deciduous (*Vaccinium myrtilloides*) shrubs. The shrubs have an average height
of 0.2-0.3 m and leaf area index (LAI) of ~ 1.3 (Moore et al., 2002). Both hummocks and hollows are covered by
a continuous mat of Sphagnum mosses (*S. capillifolium, S. magellanicum*), with the capitulum area cover (LAI)
is ~ 1.0. Total aboveground and belowground biomass for vascular plants, measured in 1999 averaged $356 \pm 100$

g m$^{-2}$ and $1820 \pm 660$ g m$^{-2}$, respectively, and Sphagnum capitulum biomass was $144 \pm 30$ g m$^{-2}$ (Moore et al.,
2002).

The beaver ponds on the southern edge of the peatland are ~ 250 m from the water table loggers at the peatland
flux tower site (Fig. 1). Energy, water, and $CO_2$ fluxes have been continuously measured at Mer Bleue over the
last 20 years. Roulet et al. (2007) provide an overview of measurements of C exchanges of the site including $CO_2$,

$CH_4$ and dissolved C. Fraser et al. (2001a) presented the details of the peatland groundwater flow and runoff
hydrology. For details of vegetation composition and biomass measurements, see Moore et al. (2002) and Bubier
et al. (2006).

**2.2 Beaver pond water level measurement and disturbance**

The water level at the BP was measured *circa* weekly from April to November using manual stage height

recordings, starting from 2004 to 2019 (Fig. 2). Fraser et al. (2001a) measured the water level at the BP (i.e., $P_1$
well) for a complete year, 1998-1999. Unfortunately, only periodic measurements were taken during 1999-2003,
when the beavers were removed and the beaver dams were not maintained, so we generated a random water level
series based on the mean and standard deviation value of the available BP water level over 1999 to 2003. BP water
level under different assumptions over this period were assessed in Fig. S1 (Supplement Section A).

Two additional beaver dam disturbances occurred at Mer Bleue: one in the fall of 2012 and the other in May
2018. A beaver baffler, a long perforated PVC pipe, was installed through the beaver dam in the fall of 2012 in
an attempt to implement a controlled experiment to lower the BP water level. However, the experiment failed due
to the industriousness of the beavers. At a local scale, Goud et al. (2017) conducted a study during 2012-2013
using the time window when the water level briefly lowered between periods after the researcher cleared the pipes

and the beavers blocked them. In 2018 the beaver dam was partially opened by the NCC to alleviate water pooling
at a nearby road and repeated blockage of a road culvert, resulting in the partial drainage of the BP during May
2018, but after ~ 1 month renewed beaver activity returned the BP to the pre-drainage water levels.

**2.3 Brief model description**





The CoupModel platform (coupled heat and mass transfer model for soil–plant–atmosphere systems) is a process-based model designed to simulate water and heat fluxes, along with the C cycle, in terrestrial ecosystems (Jansson, 2012). The main structure is a one-dimensional, vertical model, with one or two layers of vegetation (for example a shrub and moss layer as in this application) on a multi-layered soil profile. In this application, we used the most recent model version (v.6) from He et al. (2021), but without nitrogen and phosphorus cycles. The model is driven by measured climatic data – precipitation, air temperature, relative humidity, wind speed, and global radiation. Vegetation is described using the "multiple-big leaves" concept with two vegetation layers (shrubs and mosses) simulated taking into account mutual competition for light interception and water uptake (Jansson and Karlberg, 2011). The model and technical description are available at www.coupmodel.com. The model structure and process description of CoupModel used for fen peatland conditions are described in Metzger et al. (2015). The key model concepts, surface energy fluxes and partitioning (eq. S1-2), evapotranspiration and aerodynamic resistance (eq. S3-5), plant water uptake (eq. S6), soil heat (eq. S7), and photosynthesis and respiration (eq. S8-9) and their parameterizations are described in detail in Supplement Sections B and C. Below, we only describe the setup relevant to Mer Bleue bog, which is lateral flow to the BP and peat hydrology.

### 2.4 Simulation design, initial and boundary conditions

CoupModel was used to simulate the C balance and linked hydrology, energy fluxes of Mer Bleue at hourly resolution from September 1, 1998 to January 1, 2020.

We divided the peat soil profile into 16 layers: from 0.04 m per layer in the top to 0.60 m per layer in the bottom. Overall, 3.4 m peat was simulated with eight layers for the acrotelm (peat surface to ~ 0.4 m peat depth) and eight layers for the catotelm. The initial conditions for the plant cover and its compartments were derived from measured data (Moore et al., 2002). The model conceptually divides the shrubs into leaf, stem, coarse root, and fine root. For Sphagnum mosses, the capitulum was conceptually viewed as leaf and the rest as stem in the model. Initial conditions for soil C were taken from measured soil profile data (Blodau and Moore, 2002; Frolking et al., 2002; Dimitrov et al., 2010). This study used three soil organic matter pools, which differed in substrate quality and decomposition rate (Table S1 in Supplement Section C), to simulate the soil C dynamics: the shrub litter, moss litter, and refractory organic matter. Decomposition of the shrub and moss litter would add to the refractory organic matter, while the decomposition of refractory organic matter only released $CO_2$ into the atmosphere. As in Metzger et al. (2015), we partitioned the initial soil pools by Mer Bleue measured C/N ratio as an indicator of the organic matter quality. The measured C/N ratio data of plant litter was 75 for shrub litter, 55 for moss litter, and 30 for the refractory organic matter (Moore and Bubier, 2019; Wang et al., 2014).

Measurements show a mineral substrate beneath Mer Bleue with very low hydraulic conductivity $\ll 10^{-10}$ m s$^{-1}$ (Fraser et al., 2001b) so we assumed a no-flow boundary at 3.4 m depth. A geothermal heat flow was assumed for the lower boundary condition. To account for the hummock-hollow microtopography, we adopted the parameter upscaling approach (Wu et al., 2011). This means the model did not account for the microtopography explicitly but used the integrated parameter values that weighted for the percentage coverage of hummock and hollow instead. The model parameter values were mostly obtained from Mer Bleue measured data, and if not, literature values from previous model applications were used (Table S1). To account for the continuous accumulation of peat, the annual organic matter increases in all the layers (except the last soil layer) was moved from one layer to the layer below at the end of each year. The amount moved is the annual accumulation of the peat C, similar to





the Clymo (1984) model. This results in a constant peat surface, which is the reference elevation for the soil physical property, hydrology calculations, and the surface – atmospheric exchanges. In other words, the peat
increases in mass without an actual change in physical height.

### 2.5 Model description of hydrology and lateral flow to the beaver pond

Water flow in the peat soil was assumed to be laminar and thus follows Darcy's law as generalized for unsaturated flow by the 1-D Richards's equation (Richards, 1931). The change of unsaturated conductivity with water content was modeled following Mualem (1976). The relationship between water tension and soil water content was
approximated by the van Genuchten (1980) approach. Coefficients of the van Genuchten function (Table S1) were obtained from the synthesized empirical data of peat soil from Weiss et al. (1998) and Letts et al. (2000), with the residual water content set to 0% and the wilting point to 10% (vol.) for all the layers (Schwärzel et al., 2006). The vertical saturated hydraulic conductivity profile was derived from the mean of the measured data of Fraser et al. (2001b). The horizontal hydraulic conductivity, $k_{sat}$ for acrotelm ranged from $10^{-7}$ to $10^{-3}$ m s$^{-1}$ and increased
toward the peat surface. The $k_{sat}$ for catotelm peat ranged from $10^{-8}$ to $10^{-6}$ m s$^{-1}$ (eq.1, Table S1).

Fraser et al. (2001a) studied the groundwater flow in Mer Bleue. They suggested that groundwater flowed perpendicularly to the BP and can be reasonably described by the Dupuit-Forchheimer assumptions. Thus, in our study the BP was conceptually viewed as a dynamic stressor: an open channel with a varying water level into which the lateral flow in Mer Bleue drains. The lateral subsurface flow (seepage flow) was assumed to be driven
by the hydraulic gradient of the water level at the peatland tower site and the BP. Water flow occurred when the simulated groundwater table in the peatlands, $z_{sat}$, was above the water table level in the BP, with flow occurring horizontally from a peat layer to BP when the soil was saturated (eq.1). The subsurface flow rate, $q_{sub}$ was assumed proportional to the hydraulic gradient ($z_{sat}$ - $z_{bp}$)/$d_p$, the thickness, $d_z$, and saturated hydraulic conductivity of each soil layer, $k_{sat}$:

$$q_{sub} = \int_{z_{bp}}^{z_{sat}} k_{sat} \frac{(z_{sat} - z_{bp})}{d_u d_p} dz \qquad\qquad (eq.1)$$

The $d_u$ was the unit length of the horizontal element – i.e., 1 m. $z_{bp}$ was the water level at the BP.

During over-saturated periods, such as snowmelt and heavy rainfall, the flow of water in the upper soil layer can be directed upwards and added to the surface runoff to the BP. Briefly, the model formed a pool of water on the soil surface, when the throughfall exceeded the infiltration capacity of the first soil layer. Water in the surface
pool could either infiltrate with a delay into the soil profile or be lost as surface runoff, as overland flow.

### 2.6 Data used for model evaluation

The model was evaluated against hydrological and $CO_2$ flux data from Mer Bleue. Due to the strong coupling of hydrological components to energy fluxes in the CoupModel, we further evaluated the model output against observed energy fluxes. Overall, the dataset used for evaluation consisted: 1) hydrological variables including
evapotranspiration measured by the eddy covariance system (EC) (1998-2018), water table depth (1998-2019), snow depth (1998-2014), and total runoff measured at the outlet of the catchment (2011-2014); 2) $CO_2$ fluxes (1998-2018) including gap-filled NEE, derived gross primary production (GPP), and ecosystem respiration (ER); and 3) energy flux data (1998-2018) including radiation fluxes, turbulent energy fluxes, and peat soil temperature



profile. Details of NEE exchange measurements and data processing were described in Lafleur et al. (2001; 2003) and Roulet et al. (2007). The evaluation was conducted using time series and goodness of fit quantified by the linear regression coefficient of determination ($R^2$) and mean values of simulated and measured data.

## 3 Results

### 3.1 Lateral flow to the beaver pond and hydrological fluxes

The measured BP water level ranged from 68.2 to 69.4 AMSL over the 21 years, where beaver trapping and dam
breaking altered the pond water level significantly (Fig. 2). We classified the BP water level into three distinct periods according to the measured change in water level: 1) high disturbance period (1998-2004), where water levels were ~ 1 m lower than the present-day average (2017-2019); 2) moderate disturbance period (2012-2016) with water level ~ 0.30 m lower than the present; and 3) no disturbance period over 2004-2012 and 2017-2019 when higher water levels were maintained by an intact beaver dam. As described above, a disturbance in May
2018 lowered the water level by < 0.1 m, but the water levels quickly returned to pre-drainage levels in ~1 month (Fig. 2). In this study, we included the May 2018 event in the 'no disturbance' period.

The hydraulic head difference between the peatland and the BP, a distance of 250 m (Fig. 1), ranged from 1.2 to ~0 m over 1998-2019 (Fig. 2), which gave a lateral hydraulic gradient, from 0.0045 to ~0. Using the measured vertical saturated hydraulic conductivity from Fraser et al. (2001a), the hydraulic gradient (eq. 1) led to a simulated
mean annual integrated subsurface flow of $327 \pm 162$ mm yr$^{-1}$, ~ one-third of annual precipitation over 1998 to 2019. Flow mainly occurred during the spring, autumn, and winter seasons (Fig. 3b). The ratio of annual subsurface flow to precipitation was $0.4 \pm 0.1$, $0.23 \pm 0.06$, and $0.35 \pm 0.1$ during the high, moderate, and no disturbance periods, respectively. The ratio during high disturbance was significantly greater ($p < 0.0001$) than for the other two periods, which were not significantly different.

Simulated evapotranspiration agreed well with the measured data (coefficient of determination, $R^2 = 0.73$) (Fig. 3a, b). Over 1998-2018, the simulated average evapotranspiration was $430 \pm 34$ mm yr$^{-1}$, which was similar to the measured $490 \pm 42$ mm yr$^{-1}$ (Fig. 3a, b; Fig. S3a in Supplement Section D). Using the measured data, the ratio of evapotranspiration/precipitation was $0.50 \pm 0.09$, while the simulated value was $0.44 \pm 0.08$. Disturbance level did not show a clear impact on the ratio.

Simulated water table depth (WTD) in the peatland showed reasonable agreement with the measured data (Fig. 3c; Fig. S3c). The $R^2$ values were 0.33, 0.40, and 0.50 for the high, moderate, and no disturbance periods, respectively, thus there was slightly better agreement with decreasing disturbance levels. The model simulated a ~ 0.05 m shallower WTD than the measured over the growing season (Fig. S3c), which was probably caused by underestimating evapotranspiration during the same period (Fig. S3a). The mean measured WTD was -0.44 ±
0.04 m (simulated: -0.45 ± 0.05 m), -0.36 ± 0.06 m (simulated: -0.38 ± 0.09 m), -0.32 ± 0.05 m (simulated: -0.32 ± 0.07 m) during the high, moderate, and no disturbance periods, respectively. The peatland WTD during the high disturbance period was ~ 0.1 m deeper ($p < 0.0001$) than for the other two periods. Overall, a higher disturbance level led to deeper WTDs in both the measured and simulated data.

The model generally overestimated the snow depth up to 50% compared to the measured data (Fig. S3b). The
overestimation also partly contributed to the underestimation in the simulated WTD during the winter periods (Fig. S3c). Over the snow-melting season (Fig. 3b), the model predicted an average surface runoff rate of 220 ±





102 mm yr$^{-1}$. This probably represents an overestimation due to a simulated larger snow depth. The simulated total runoff, combining surface runoff and subsurface lateral flow, was 545 ± 205 mm yr$^{-1}$, similar to 462 ± 163 mm yr$^{-1}$, the annual total runoff measured at the outlet of the catchment (2011-2014) (Fig. 3a). The measured and
simulated total runoff/precipitation ratio was 0.54 ± 0.07 and 0.61 ± 0.09, respectively. The simulated ratio of subsurface flow over total runoff was 0.69 ± 0.1, 0.45 ± 0.08, and 0.6 ± 0.07 during the high, moderate, and no disturbance periods, respectively. The ratio during the high disturbance period was greater ($p < 0.0001$) than for the other two periods.

### 3.2 Carbon fluxes

The simulated $CO_2$ fluxes including GPP, ER and NEE reproduced the measured data well, with $R^2$ values 0.73, 0.91 and 0.51, respectively (Fig. 4). The lower $R^2$ of NEE is due to it being the net result of GPP and ER, so carries the errors of both. The measured annual GPP, ER and NEE over 1998-2018 were, -658 ± 103, 568 ± 80 g C m$^{-2}$ yr$^{-1}$ and -90 ± 39 (negative means net $CO_2$ uptake by the ecosystem), respectively. The corresponding simulated fluxes were -670 ± 46, 599 ± 74 g C m$^{-2}$ yr$^{-1}$, -71 ± 61.

The model simulation showed that the shrubs dominated the GPP uptake, ~ 80% of the total GPP, and the mosses contributed ~ 20%. Over 1998-2019, the simulated average net primary production (NPP) was 272 ± 30 g C m$^{-2}$ yr$^{-1}$ and 82 ± 9 g C m$^{-2}$ yr$^{-1}$ for the shrubs and mosses, respectively. Thus, mosses contributed 23% of the total NPP. The simulated heterotrophic respiration and soil respiration (i.e., heterotrophic respiration and autotrophic respiration of shrub roots) were 283 ± 69 and 476 ± 76 g C m$^{-2}$ yr$^{-1}$, respectively. The heterotrophic peat respiration
thus accounted for 59% of the soil respiration or 49% of total ER. This agrees well with the available measurements from Mer Bleue showing heterotrophic respiration accounted for 30-63% of the total ER (Stewart, 2006; Rankin et al., 2021).

The model-data agreement for NEE was better during the mid-to-late-summer, autumn, and winter seasons than in the spring and early summer (Fig. 4c), probably due to model uncertainty in representing the shrub phenology.
In spring (April and May), the model generally showed a larger uptake than that measured by the EC (Fig. 4c), caused either by the earlier and stronger recovery of photosynthesis in the model (Fig. 4b) or by a slower increase of modeled respiration (Fig. 4c). The current ER underestimation (Fig. 4b) can be partly explained by the simulated shallower WTD (Fig. S3c), despite a slight overestimation of soil temperature (Fig. S4; Fig. S3d).

The variations in BP water levels showed clear impacts on the peatland $CO_2$ uptake and the bog mean WTD during
the growing season (May to October) both in the measured and simulated data (Fig. 5). The measured mean growing season WTD became shallower with decreasing disturbance level, ranging from 0.44 ± 0.05 m (simulated: 0.45 ±0.08 m), 0.41 ± 0.06 m (simulated: 0.42 ± 0.04 m), and 0.38 ± 0.04 m (simulated: 0.33 ± 0.03 m) for the high, moderate, and no disturbance periods, respectively (Fig. 5a). Accordingly, the measured mean NEE generally decreased (i.e., greater net uptake of $CO_2$) from -44 ± 45 g C m$^{-2}$ yr$^{-1}$ (simulated: -55 ± 89), -102 ± 40 g
C m$^{-2}$ yr$^{-1}$ (simulated: -67 ± 51), and -115 ± 33 g C m$^{-2}$ yr$^{-1}$ (simulated: -90 ± 35) with decreasing disturbance level (Fig. 5b).

### 3.3 Surface energy fluxes and partitioning

CoupModel reproduced the measured seasonal pattern and the magnitude of the radiation fluxes (R$_{n,tot}$, LW$_{in}$, LW$_{out}$) very well (Fig. S1a, b, c). However, the model underestimated the sensible flux (*H*) by 10-20% over the



year, except spring (Fig. S1d), and showed a 30% underestimation of latent heat (LE) during the peak growing season with slightly overestimated winter LE (Fig. S1e). Underestimation of the turbulence fluxes further explained the higher simulated soil temperature in the peat soil profile (Fig. S3d, Fig. S4). The measured and simulated Bowen ratio (B=H/LE) showed high agreement over the growing season, both ranging from ~1 to ~0.5. However, relatively high biases were found in the spring and winter seasons, probably because the biases in

simulating the snow further influence the energy partitioning during the frozen period. Interestingly, CoupModel simulated greater evaporation over winter than was observed using the EC technique (Fig. S3a). In the model, there was no photosynthesis activity of shrubs during the winter, but mosses still grew when the air temperature was above 0 °C since the model did not account for burial of moss by snow. The winter LE bias could be due to the simulation of higher evaporation from the moss layer given the high availability of water, with a thicker than

measured snow cover on the soil surface (Fig. S3b).

**3.4 Impacts of beaver pond water level variation on peatland $CO_2$ fluxes**

The validated model (hereinafter referred to as the 'reference run') was then used to analyze the influence of variations in the BP water levels on the peatland $CO_2$ exchange. First, we addressed what the peatland net $CO_2$ uptake would be if there were no variations in BP water level. In this experiment, we maintained a constant BP

water level, 68.9 AMSL – the mean level for 1998-2019 (Fig. 2), for the simulation hereinafter referred to as constant BP water level run. The simulated annual net $CO_2$ uptake over 1998-2019 was ~ 30 g C m$^{-2}$ yr$^{-1}$ higher than the reference run (Fig. 6), despite the simulated average peatland WTD being identical for both runs. Smoothing out the variation of BP water level reduced the annual variation of the simulated NEE (Fig. 6, box plot), which, in turn, suggests its influence on NEE. For the high disturbance period, the constant BP water level

run predicted a ~ 0.1 m shallower peatland WTD and an average 55 g C m$^{-2}$ yr$^{-1}$ higher net $CO_2$ uptake, compared to the reference run (Fig. 6). Given that the measured NEE uptake was 44 ± 45 g C m$^{-2}$ yr$^{-1}$ (Fig. 5), the constant BP water level run suggests the NEE uptake within this period would have been ~ 99 g C m$^{-2}$ yr$^{-1}$ without the disturbance. This updated number was close to the measured 21-year average of C uptake in Mer Bleue, 90 ± 39 g C m$^{-2}$ yr$^{-1}$ (110 ± 37 for 2005-2018).

We conducted a sensitivity analysis of the BP water level, i.e., varying boundary conditions for the hydrology setting of the system, on the simulated $CO_2$ fluxes. For this analysis, the BP water level was kept constant throughout each run, but there was still variability in the peatland WTD due to differences in inputs and outputs, though the gradient between the peatland WTD and BP was constant through each simulation. The BP water level was varied from 67.9 AMSL (a dry pond) to the highest possible water level, 69.37 AMSL (the average measured

WTD at the peatland) when flow would reverse from the pond to the peatland. This range in BP water level changed the lateral hydraulic gradient between 0.005 and ~ 0.0005. The $CO_2$ uptake showed a close to linear relation with the lateral hydraulic gradient and the BP water level (67.9-69.2 AMSL) (Fig. 7). The slope of the linear regression was ~ -120 g C m$^{-2}$ yr$^{-1}$ m$^{-1}$. Thus, a one-meter drop in BP water level would lower the average peatland WTD by ~ 0.15 m, and this produced a ~ 120 g C m$^{-2}$ yr$^{-1}$ reduction of the simulated peatland net $CO_2$

uptake (Fig. 7). The decreased net $CO_2$ uptake was attributed to a greater increase in ER than GPP with a decrease in BP water level (Fig. 7). However, the linear relationship breaks down when the BP water level gets closer to the WTD in the peatland (69.2-69.4 AMSL, Fig. 2; Fig. 3c).





Overall, Fig. 7 also shows that the current model could simulate the complex self-regulating behavior of the bog. The model also suggests that there is a tipping point around a BP water level of ~ 68.1 AMSL, or 1.7 m below the

average peat surface, which represents the lower end of the self-regulation for the bog system. At water levels below this tipping point, NEE becomes positive. This means that to ensure the Mer Bleue peatland continues to function as a C sink, the beaver pond, the boundary for hydrological setting, must operate in the threshold range of above ~ 68.1 AMSL. Within this range, the bog's net $CO_2$ uptake is resilient to variations in BP water levels.

## 4 Discussion

### 4.1 Model evaluation for the continental bog system

Current model evaluation with a detailed dataset from the Mer Bleue peatland shows CoupModel can simulate the ecosystem energy, water and $CO_2$ fluxes well with varying hydrological boundary conditions. However, the results show the CoupModel simulates a higher NEE uptake in the spring than was observed (Fig. 4c). We believe this bias most likely originates from the poor representation of phenology of the evergreen shrubs (i.e.,

*Chamaedaphne calyculata*). In the model, the shrub phenology (onset and end of photosynthesis) is calculated based on a simple air temperature and degree day function. Once the growing season starts in the model, full photosynthetic capacity, regulated by light, air temperature, soil water (eq. S6) is assumed. However, data from the onsite phenology camera and satellite-based products all show a gradual spring recovery, from April to May, of the shrub's greenness (Kross et al., 2014; Arroyo-Mora et al., 2018). Previous studies also show spring recovery

of the shrubs is dependent on their roots becoming thawed and recovery of leaf nitrogen and chlorophyll concentrations (Moore et al., 2006). Imposing a more gradual shrub greening of 25% greenness in April to 100% by late May (Sonnentag et al., 2007), reduces simulated spring $CO_2$ uptake and reduces annual GPP by ~ 10% (data not shown). The relationship between peatland NEE and the lateral hydraulic gradient between the BP and the peatland still exists regardless of spring phenology, since the annual NEE is dominated by summer and autumn

uptake at Mer Bleue (Lafleur et al., 2003) with a relatively small contribution from the spring recovery (Moore et al., 2006). However, the model evaluation highlights the potential importance of spring phenology and indicates a need to refine the current algorithm describing when and how the rate of plant production increases, particularly if CoupModel were to be used to simulate the effect of climate change on Mer Bleue or similar continental bog systems. Metzger et al. (2015) evaluated CoupModel with multiple peat sites in Europe and found different

descriptions of plant phenology were not needed to simulate peatland $CO_2$ dynamics. The differences in the sensitivity of NEE to the parameterization of spring phenology in our study versus Metzger et al. (2015) may arise because Metzger et al. examined poor fens, or a restored bog with high nutrient conditions with a moderated climate, both of which differ from conditions at Mer Bleue. The continental climate at Mer Bleue with cold winters and warm summers might further explain the more pronounced phenology compared to the European sites.

Our evaluation shows that the plant biomass (data not shown) of Mer Bleue is quite resilient to changes in hydrological conditions. At the lowest simulated BP water level (67.9 AMSL, WTD to ~ -0.6 m, Fig. 7), the LAI of the shrubs only increases 15%, while the mosses decrease by 4%. More than 20 years of observation in a bog in Connecticut, USA showed little change in the bog shrub and moss biomass under changing hydrological conditions (Mitchell and Niering, 1993; Mcmaster and Mcmaster, 2001). A 15- year modest drawdown of the





water table in a peatland complex of southern Finland also shows a small effect on bog vegetation cover, compared to the larger changes (increase in vascular cover) in poor and rich fen sites (Kokkonen et al., 2019).

**4.2 Peatland C fluxes response to hydrological boundary changes and implications**

Our study shows that variations of the water level in the BP adjacent to a peatland can influence the peatland's WTD 250 m away, primarily by altering the lateral hydraulic gradient and the subsurface flow. Lafleur et al. (2005)
analyzed the evapotranspiration data from Mer Bleue and found that the maximum rooting depth of the shrubs (~0.65 m) is a critical control of the evapotranspiration. At WTD above -0.65 m mean evapotranspiration did not change with decreasing WTD. In this study, mean WTD analyzed under various disturbance levels ranged from -0.3 to -0.6 m and did not result in significant changes of evapotranspiration (Fig. 3).

Fraser et al. (2001b) measured the total runoff and the runoff from snowmelt (corresponding to the modeled
surface runoff) at the catchment outlet over 1998-1999, and showed that snowmelt accounted for ~53% of the total runoff. The simulation generated a similar ratio of snowmelt runoff to total runoff, 60%. This agrees with observations from many other boreal peatlands that runoff response under high flow rate conditions, such as snowmelt, are rapid (Holden, 2005; Verry et al., 1988). CoupModel estimates one-third of the precipitation was drained to the BP in the studied period. This might represent a higher end of the lateral subsurface flow due to the
BP disturbance at Mer Bleue. Assuming our subsurface lateral flow/total runoff ratio during the no BP disturbance period (~ 50%) applies for the pristine bog conditions of northern peatlands and given the total runoff in those bogs normally ranges 120-500 mm yr$^{-1}$ (Fraser et al., 2001b), annual water flow to the adjacent BP under no disturbance conditions accounts for ~ 60-250 mm yr$^{-1}$. Therefore, the BP plays an important role in regulating the hydrology of Mer Bleue and probably other northern peatlands (Westbrook et al., 2020; Woo and Waddington,
360    1990).

Variations in BP water levels are shown to influence the net $CO_2$ uptake of Mer Bleue. The effect on $CO_2$ fluxes is dependent on the intensity and direction of the BP water level variation (Figs. 5 and 7). Goud et al. (2017) measured the $CO_2$ fluxes at a local scale over a transect from the hummock, hollow, bog margin to pond margin (~ 50-100 m away from the BP) of Mer Bleue during the moderate disturbance period, and found that the measured
GPP, ER from the four microsites were all reduced by half and the $CO_2$ uptake at the bog margin and pond margin were significantly reduced after the BP disturbance, which corroborates our findings at the ecosystem scale. The high BP disturbance has greater influence in regulating NEE than climate variations, as demonstrated by model simulations showing $CO_2$ uptake of the bog is reduced by 55 g C m$^{-2}$ yr$^{-1}$ when a constant BP water level was assumed (Fig. 5c), which was greater than the observed annual variations in NEE, 37 g C m$^{-2}$ yr$^{-1}$, derived from
the 14-year eddy covariance data (2005-2018). The heightened role of disturbance is also reflected by the weak correlations between NEE and measured bioclimatic variables (data not shown). But the annual NEE shows a significant negative correlation with the mean WTD over the growing season (Fig. 8). Similar control of WTD on NEE variability between years was also found for a boreal ombrotrophic bog in the James Bay region (Strachan et al., 2015). Humphreys et al. (2014) conducted two-year NEE flux measurements at two ombrotrophic bogs in
Hudson Bay lowlands, where the annual flux is closely linked to the lowering of WTD during summer. Recently, Zhang et al. (2020) reconstructed the historical peatland C uptake over a millennial-scale of two dwarf shrub bogs in Finland and found a strong correlation between the C uptake and water conditions. Hence, WTD is likely a significant driver for NEE fluxes in boreal bogs. Lower water tables due to climate change are predicted for boreal



peatlands. Currently, peatland models generally predict a reduced $CO_2$ sink strength for boreal bogs by the end of

the 21st Century (Qiu et al., 2020). Wu and Roulet (2014) predicted the $CO_2$ uptake of the Mer Bleue bog will decrease by 16-45 g C m$^{-2}$ yr$^{-1}$ compared to current climate reference runs. The long-term C accumulation rate for boreal peatlands is estimated to be 10-50 g C m$^{-2}$ yr$^{-1}$ (Loisel et al., 2014), and C accumulation rate is predicted to reduce by ~27 g C m$^{-2}$ yr$^{-1}$ for the RCP 8.5 scenario and ~11 g C m$^{-2}$ yr$^{-1}$ for the RCP 2.6 scenario (Chaudhary et al., 2017). Simulations of peatlands with climate change do not yet capture the self-regulating aspect of peatland

C dynamics. Our simulations show that the variations in water levels connected to BP disturbance could potentially influence the peatland $CO_2$ uptake of a similar magnitude or even greater than that projected in climate change scenarios.

Theoretical studies have argued that bogs are complex adaptive systems based on the tight feedbacks among plant production, decomposition, and water storage represented by WTD (Eppinga et al., 2009; Frolking et al., 2010;

Morris et al., 2012). While the relationship between water storage and peat accumulation is well established, all the studies referred to deal with the role of self-regulation. In our study, we highlight the role that the adjacent boundary conditions might play. The empirical observations and our simulations suggest links between water storage in a peatland, the stresses to which the lateral flow responds, and the ecosystem level NEE. Water level changes due to land-use changes adjacent to peatlands can have a similar effect on changing lateral hydraulic

gradient and the peatland NEE. Studies have emphasized the hydrological role of laggs surrounding bogs to bog ecology and function (Howie and Meerveld, 2011). Others have shown the impact of bog hydrology through large regional water table drawdown (Regan et al., 2019).

The relationship between beaver ponds and bog C dynamics has potentially much broader implications as the beaver populations in North America have substantially increased over the last 100 years (Naiman et al., 1988;

Nisbet, 1989) and beaver populations are growing in Europe as well (Halley et al., 2012). With climate change, the beaver habitat and species density may further increase (Jarema et al., 2009; Tape et al., 2018). Previous studies have shown that beaver ponds are sources of $CH_4$ and $CO_2$ emissions (Roulet et al., 1997; Dinsmore et al., 2009; Huttunen et al., 2002). A rising water level associated with the increased beaver population will increase the area of the ponds as well as the emissions from the ponds to the atmosphere. However, we also expect a rising

water level at the ponds will decrease the lateral hydraulic gradient to adjacent bogs, increasing bog wetness, thus increasing net $CO_2$ uptake and $CH_4$ emissions from the bog (Moore et al., 2011). Colonization of beavers will typically increase the water level at the ponds by ~1 m, e.g. (Rosell et al., 2005; Karran et al., 2017), increasing the bog NEE uptake by 60-160 g C m$^{-2}$ yr$^{-1}$ (Fig. 7) in our study and based on seasonal WTD-$CH_4$ relationships at Mer Bleue (Moore et al., 2011), the increased WTD will lead to an increase of 1-3 g $CH_4$-C m$^{-2}$ yr$^{-1}$ emissions

at the bog. Methane emissions from the beaver ponds range from 1 to 57 g $CH_4$-C m$^{-2}$ yr$^{-1}$, with an average of ~ 15 g $CH_4$-C m$^{-2}$ yr$^{-1}$ (Whitfield et al., 2015) and the $CO_2$ emissions between 20 and 200 g C m$^{-2}$ yr$^{-1}$, average ~ 100 g $CO_2$-C m$^{-2}$ yr$^{-1}$ (Roulet et al., 1997; Huttunen et al., 2002). We approximate the peatland and pond areas by assuming a rectangle with length from the drainage divide in the center of the Mer Bleue and the width of the BP (Fraser et al., 2001a). This gives the percentage cover of the pond over the adjacent peatland bog from 2% to 10%.

At the landscape level, the total increased emissions based on C mass due to beaver pond ($CH_4$ from pond and bog, $CO_2$ from pond) offset 4 to 12% of the increased net $CO_2$ sink of the adjacent bog. Applying the 100-year global warming potential as an example (1 g $CH_4$ = 28 g $CO_2$ equivalent, Myhre et al. (2013)), the total increased emissions in terms of $CO_{2\ equivalent}$ will offset 26 to 42% of the increased C sink of the bog. Although the current



calculation has a modest uncertainty, the key message is that even in the extreme-case scenario, with maximum
range and using global warming potential rather than radiative forcing (Frolking et al., 2006), the increased bog
$CO_2$ uptake can still offset the increased $CH_4$ emissions by a considerable margin. Globally, the resurgence of
native beavers and their introduction in other regions were estimated to increase $CH_4$ emissions by 0.18-0.80 Tg
$CH_4$ $yr^{-1}$ (Whitfield et al., 2015). Our results suggest that the increased $CO_2$ sink in the bogs adjacent to BP could
largely offset the high emissions from the pond itself, thus the climate impact of beavers may be smaller than
previously thought (Nisbet, 1989; Whitfield et al., 2015).

## 5 Conclusion

This study applies the CoupModel to quantify the impact of water table variations in a beaver pond adjacent to a
mid-continental bog on ecosystem $CO_2$ dynamics. We conclude:

1) CoupModel can describe the coupled hydrology, energy and $CO_2$ dynamics for a continental bog system
430          with varying disturbance intensity induced by beaver management.
2) Beaver removal and breaking the integrity of the dams to reduce flooding decreases WT in the pond,
which increases the lateral subsurface flow from the bog lowering the peatland water table depth.
3) Peatland water table depth is a primary control of the annual NEE at Mer Bleue bog and the $CO_2$ uptake
shows a close to linear relation with the water level at the beaver pond.
4) The current modeling exercise predicts that the BP water level reaches a tipping point at ~ 1.7 m below
the peat surface where further drop in the water level would cause the bog system to no longer be resilient
to water table variations and no longer function as a net $CO_2$ sink.

*Code and data availability*. The data that support the findings of this study will be openly available in Zenodo.
The model source code used for this study is openly available in Zenodo at
https://doi.org/10.5281/zenodo.3547628. The simulation files will be openly available in Zenodo after the paper
is accepted. Much of the flux data from the Mer Bleue is available from the FLUXNET dataset (FLUXNET
Canada Research Network - Canadian Carbon Program Data Collection, 1993-2014 (ornl.gov)) and the executed
CoupModel file is available at www.coupmodel.com.

*Author contributions*. HH and NR conceptualize the work, EH, PL, TM, and NR contributed with measurement
data; HH performed the modelling and analyzed the data with help of NR; HH and NR drafted the manuscript
with all authors contributed to the paper reviewing and editing.

*Competing interests*. The authors declare that they have no conflict of interest.

*Disclaimer*. Publisher's note: Copernicus Publications remains neutral with regard to jurisdictional claims in
published maps and institutional affiliations.



*Acknowledgments.* H. He has been supported by a Canadian NSERC Discovery grant and a grant from the Trottier Institute for Science and Public Policy to N.T. Roulet. The long-term measurements at Mer Bleue have been supported by various grants to N.T. Roulet, E. Humphreys, T. Moore, and P. Lafleur (NSERC research network grants and strategic partnerships, FLUXNET Canada and the Canadian Carbon Program, BIOCAP Canada, and the Ontario Ministry of Environment and Climate Change). We thank Mike Dalva for collecting the beaver pond

data at Mer Bleue bog and the National Capital Commission for permission to work at Mer Bleue.

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



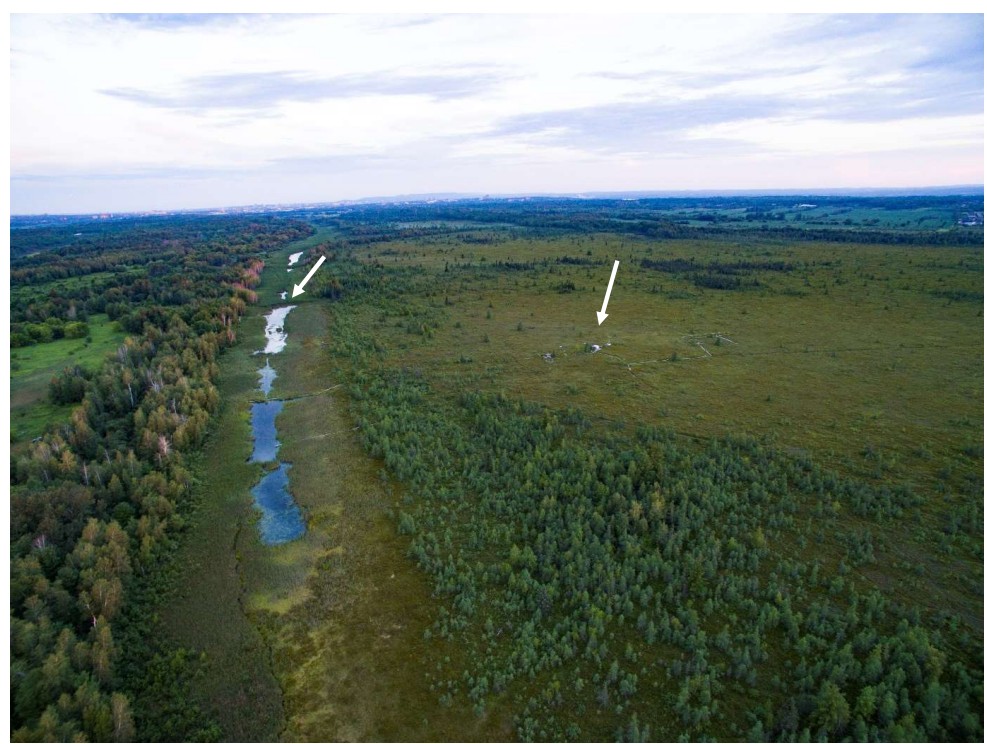

**Figure 1: Oblique photograph of Northeast portion of Mer Bleue bog. Arrows indicating the location of the beaver pond and its dam (left) and the tower (right) (Photo courtesy of Professor Margaret Kalacska, photo credit Olivier Lucanus).**






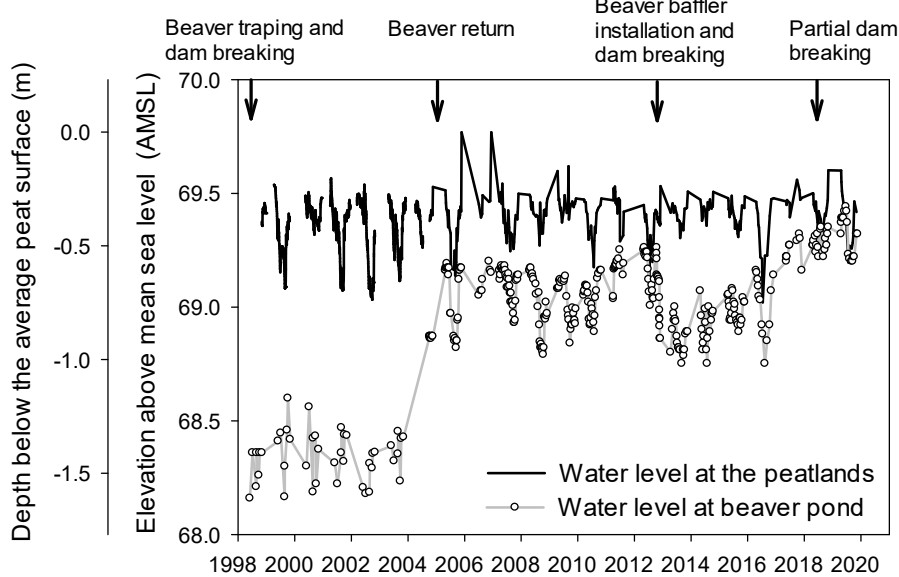


**Figure 2: Measured water level at the peatlands and the BP over 1998-2019. The arrow indicates the disturbance of the BP dam by human management (personal communication from Eva Katic, NCC).**




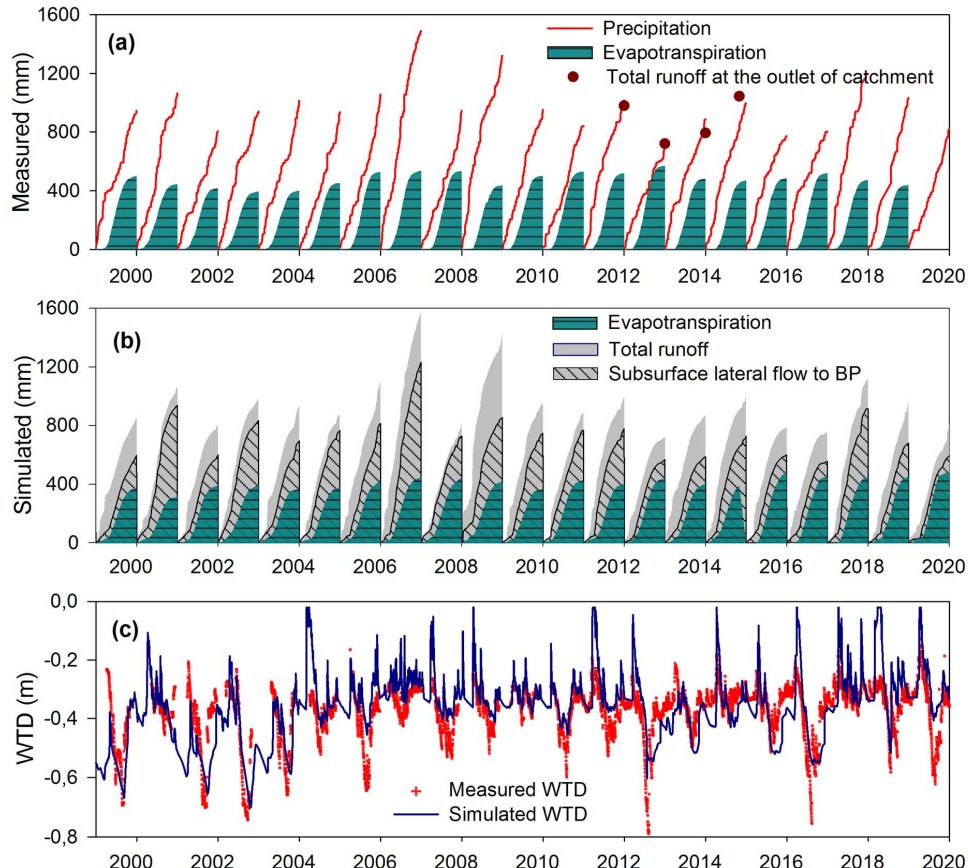

**Figure 3: (a) Simulated and (b) measured annual accumulated hydrological components: precipitation, evapotranspiration, total runoff, and the lateral subsurface flow, (c) peatland water table depth, and WTD. The measured evapotranspiration data was from the eddy covariance flux tower. Total runoff measured at the catchment outlet is taken from unpublished data Hutchins (2018).**

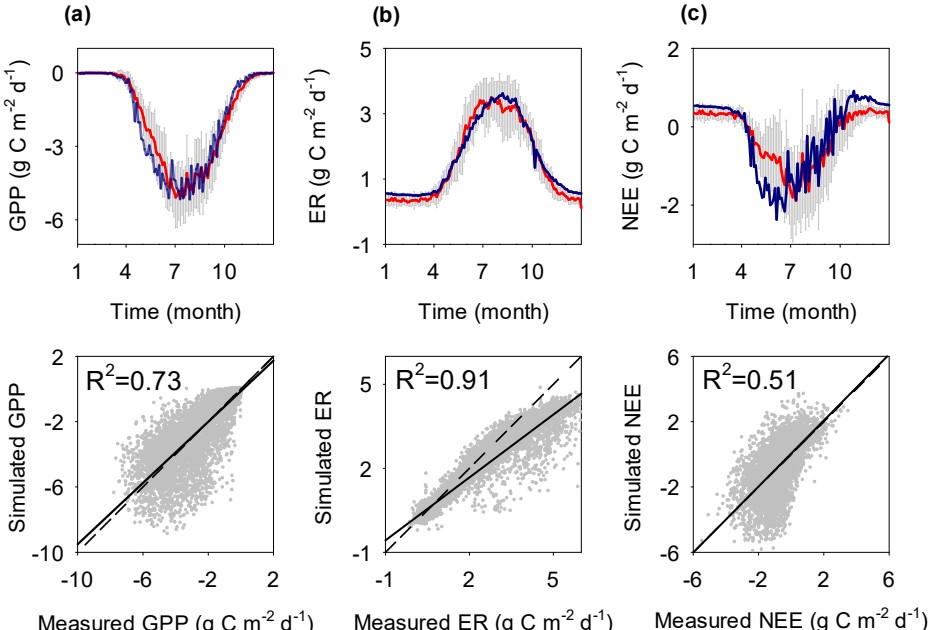

**Figure 4: Mean annual seasonal cycle of simulated (blue line) and measured (red line ± standard deviation as grey) ecosystem CO₂ fluxes, and scatter plots of simulated vs. measured fluxes: (a) gross primary production GPP, (b) ecosystem respiration ER, and (c) net ecosystem exchange NEE (1998-2018). Linear least-squares regressions (black lines) are fitted to the daily data. The 1:1 relationship is shown as a faint dotted line. R² denotes the coefficient of determination.**






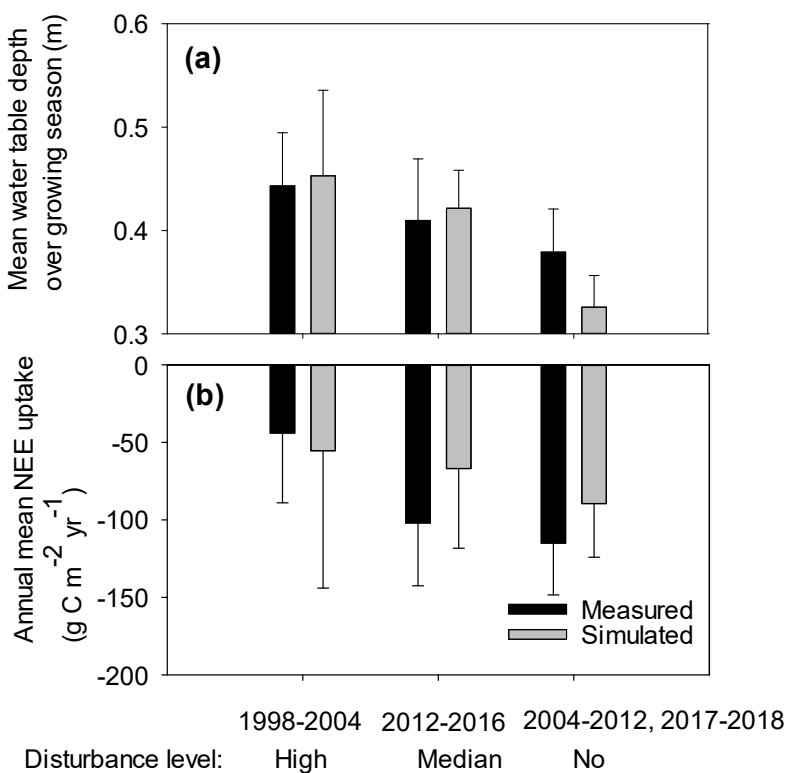

Figure 5: Simulated and measured (a) average mean water table depth over the growing season (May to October), plotted as positive and (b) annual NEE uptake for the different disturbance levels.



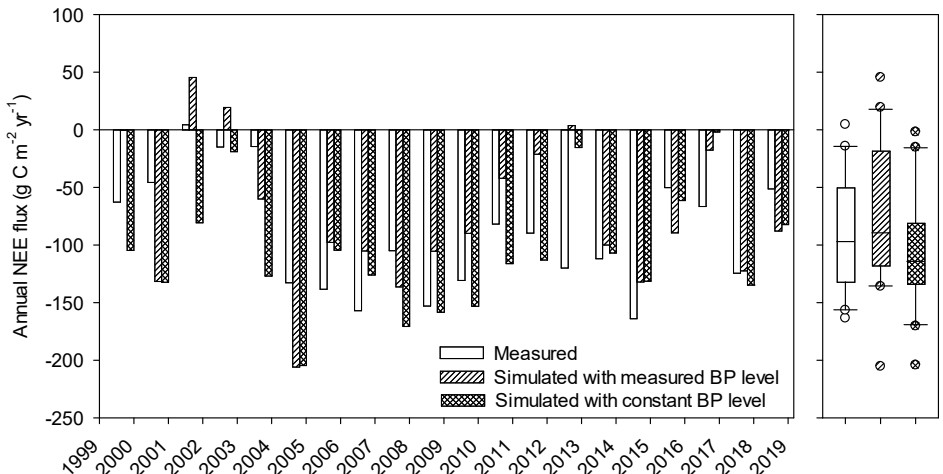

**Figure 6: Simulated and measured annual NEE fluxes in the reference run (simulated with measured BP level) and the constant BP water level run (simulated with constant BP level).**

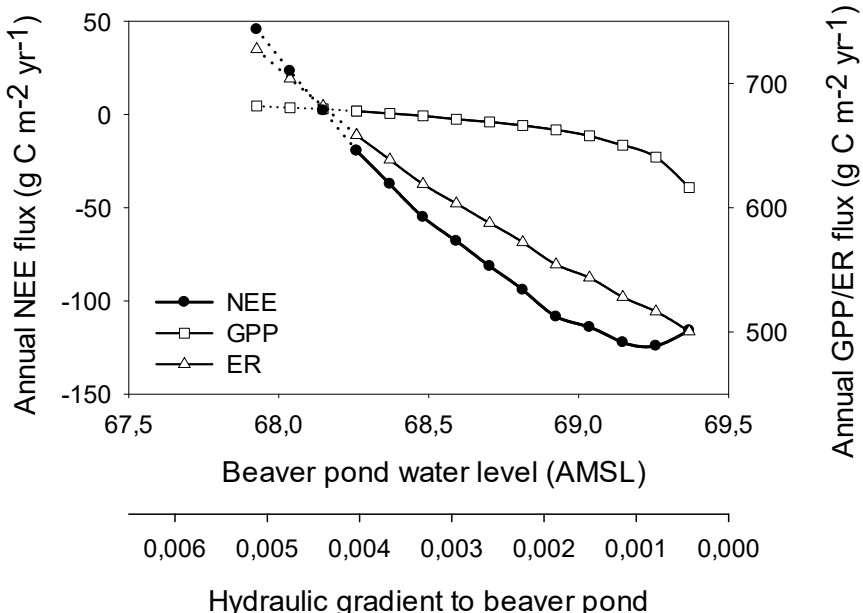

**Figure 7: Sensitivity of annual mean NEE, GPP, ER to the varying beaver pond water level (thus lateral hydraulic gradient) for each 0.1 m bin of the pond level over 1998-2019. The constant BP water level ranged from 67.9 AMSL (a dry pond) to the highest possible water level, 69.37 AMSL (the average measured WTD at the peatland) when flow would reverse from the pond to the peatland. The solid line indicates the measured BP water table range. Note the sign of NEE follows the atmospheric convention, negative meaning uptake, and GPP and ER are both plotted as positive.**





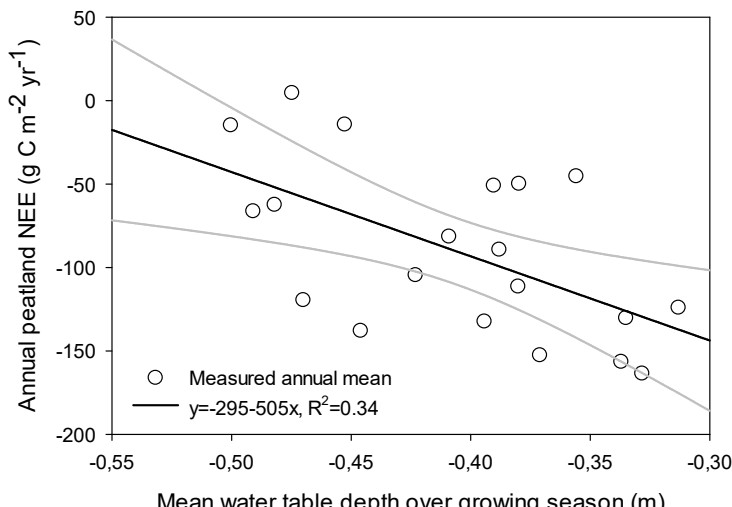


**Figure 8: Relationship between annual NEE and peatland water table depth over growing season (May to Oct) using the measured annual data 1998 to 2018.**