# Peer review of "Water level variation at a beaver pond significantly impacts net CO2 uptake of a continental bog"

_Hydrology and Earth System Sciences, 2021_

## Author Response (AR1)

**Reply to referee comment (#1 by Joshua Ratcliffe) on HESS-2021-585**

*We thank Joshua Ratcliffe for his detailed review and constructive comments. In the following, our response and list of changes during revision are given in italic, and the referee comments in normal text.*

In this manuscript the authors investigate how the management of a beaver dam has affected the lateral flow of water, and the vertical CO2 fluxes, from a well known and long running GHG monitoring site at Mer Bleue peatland. They use a modelling approach to provide several 'what-if' scenarios to model what the CO2 flux may have been without any alteration of the beaver pond level. This is a very innovative paper which addresses an important and overlooked aspect of the peatland GHG flux, i.e. what is going on at the margins. It will additionally be good to see this published as it concerns Mer Bleue, the longest running EC record on a boreal peatland and a site from which much of our understanding of boreal peatland ecology and gas flux is derived.

I really like the approach the authors have used to tease out the effect of the beaver pond from the background variability. I think it is clear from the measured data in Figure 2 and Figure 3 and Figure 6 that there was a major change in the ecosystem corresponding with the rise in the water level of the beaver pond, the coup model nicely confirms this as a mechanism.

Importantly, this confirms the wide-ranging ecological effects water table management can have hundreds of meters from the flux measurement site. This has important implications for other sites and boreal ecosystems, including other long-running flux measurement sites. I know some researchers who work on relatively degraded peatlands, with low horizontal hydraulic conductivity, who are sceptical that water level management hundreds of meters away can have any affect at all and I think it would be good to discuss how Mer Bleue might contrast to other peatlands. See my comment to this further below.

The coup model itself preforms adequately, and the authors discuss the limitations of the model in appropriate detail.

While I generally think this is a very good paper, I thought it was lacking some detail which would ensure it's comparability to other sites and situations. While hydrological feedbacks are mentioned, it's not really clear just how strong these feedbacks can be. In the discussion I would like to see a short paragraph about the horizontal hydrologic conductivity and how Mer Bleaur compares to other peatlands and how we might expect this to change if the water table was to undergo sustained lowering for a period of decades or more.

**Response**: *We agree that it would be interesting to discuss the broad implications of our results, for example "how far away does a change in the marginal stressor impact net $CO_2$ uptake of the bog". For that, we will do a back-of-the-envelope calculation to show the degree of influence as a function of distance from the disturbance. Currently, the sensitivity of hydraulic gradient to the beaver pond water level in Fig. 7 was discussed only by changing the water level, but in the revision, we will discuss the influence of distance from the $CO_2$ measurement site (drainage distance), which also, affects the lateral hydraulic gradient on the GPP, ER and NEE fluxes.*

*Moreover, in the revision, we will also conduct an uncertainty analysis to quantify how the uncertainties of the parameters in saturation conductivity (and other key model parameters, see the response to reviewer #2) influence the hydraulic feedback and C fluxes.*

**Changes during revision:** *We have added the following in the discussion section 4.2 in our revision: Line 376-380. We have added a comprehensive uncertainty analysis using GLUE methodology in the supplementary section F: Line 173-236*

One thing that bothers me rather a lot is that the increase in ER with the beaver pond level listed in Figure 7 seems to contradict what was established in Lafleur et al., 2005 (Ecosystem Respiration in a Cool Temperate Bog Depends on Peat Temperature But Not Water Table) where water table fluctuations were found to have little or no effect on ER, this is maybe due to Lafleur et al., 2005 only having the early data available, before the beaver dam raising, but I would like to see this addressed. At the moment this paper is not cited.

**Response**: *We thank the reviewer to point this out. We have investigated this further and will include discussions in our revision to make it clear. We believe the different relationships between these two studies can be explained:*

*First, Lafleur et al. (2005) study was conducted in the earlier years of the site, and in their studied five years, water table depth fell to 60 to 70 cm beneath the surface in late summer. At this depth, the peat is relatively well decomposed and changes in the water table do not mean pronounced aerobic-anaerobic transitions. Thus, a decline in the water table would not increase respiration much.*

*Second, these two studies have different temporal scales. In the Lafleur et al. (2005) paper hourly ER data (also note only nighttime data were included) were used but Fig. 7 in our study represents the average of 21 years of fluxes (ER were partitioned from NEE and thus included daytime and nighttime). In addition, the earlier study does not include the transition from low BP water levels to higher BP water levels, of which ER responses to changes in the water table more pronouncedly. Thus, a direct comparison between those two is not straightforward.*

**Changes during revision:** *We have added the following in the discussion section 4.2 in our revision: Line 384-388.*

Additionally, I have a comment about the measured data. I do not believe (nor would it be correct) that the methodology used to process the fluxes is the same as in Roulet et al., 2007. There have been several large changes in best practice for flux processing in the last 15 years that I am sure the authors are aware of. I would like to see the detailed method for flux processing included in the SI.

**Response**: *For consistency, we largely retained the flux processing methodology from the 2007 paper. The main difference with methods commonly used today and facilitated by LI-COR's*

*EddyPro software is that spectral corrections were not applied. Spectral corrections increase the magnitude of the 30 min fluxes but also add the potential for bias, particularly on shorter towers where closed path instruments and where maximization of the covariance to assess time lags are employed (e.g. Peltola et al. Atmos. Meas. Tech., 14, 5071–5088, 2021).*

**Changes during revision:** *We have added the supplementary section E to describe the method for flux processing and the potential uncertainty associated with that processing: Line 151-169.*

Could the authors state why the flux simulations in 2013 and 2017 performed so poorly compared to other years?

**Response**: *We believe the reviewer was commenting on the 2012 and 2016 years which both have measured and modeled the lowest summer WTD (Fig. 3c) but show lower uptake compared to the measured NEE data. The model simulates the drier years in the high disturbance period but show deviations for these two years, which have the highest WTD seasonal fluctuations of the 21 years. The measured NEE data suggest over these two dry years NEE was reduced but not as much in the model. We suspect this can be caused by the parameter uncertainty (e.g., vegetation, roots distribution, water uptake, etc.) which was also raised by the second reviewer.*

**Changes during revision:** *We have added a comprehensive uncertainty analysis using GLUE methodology in the supplementary section F: Line 173-236. The results show the biases of WTD partly explained the NEE over these two years, and the uncertainties in phenology representation (discussed in section 4.1) can partly explain the poor performance in these two years.*

I wish the authors the best of luck with the revisions.

Figure 2: Should be clear what is generated data and what is measured. Suggest the generated data is presented in a different colour/style The measured water level at the peatlands also looks rather spiky and a bit suspect (2006 and 2007). Please check for and remove outliers if present. It would be good to state the temporal resolution in the caption.

**Response**: *We have revised the figure caption in the revision*

**Changes during revision:** *We have added the following "Note part of the 1999-2003 is generated by using available measured data, more details see supplementary section A" to make it clear: Line 709-710*

Other than this I have some minor comments the authors may wish to consider.

L9: should be "feedbacks"

L12: consider "lateral flow of water"

L29:30 This range listed is too low, see the following. Suggest an upper range of ~200 g m-2 yr https://doi.org/10.1111/gcb.13424 https://doi.org/10.1111/j.1365-2486.2010.02378.x.

**Response**: *We have revised accordingly, and specifically g CO2-C m-2 yr-1 and added the reference*

**Changes during revision:** *Line 9, 12 and 30-31*

L54-L55: There are a few analogous studies looking at road construction and how that have raised and lowered water table levels for instance: https://doi.org/10.1007/s10021-016-0092-x

**Response**: *We have looked at the study in detail, and decided not to refer to this study, as it is more land use and land use change, differs from our study here.*

L83: Should mention that it is a downward slope (being a bog I would assume so…)

L101 Please state the total number of periodic measurements (n=?)

**Response**: *We have revised accordingly and added the measured number "(2 to 3 times during the annual summer season"*

**Changes during revision:** *Line 103*

L103: This is probably fine, given the change in mean across treatments. Please differentiate this data somehow in the plots (particularly in Figure 2)

L207: Really nice to see these numbers!

213: suggest "the disturbance level"

226: Water table is relatively meaningless over winter, not really a problem.

**Response**: *We have revised accordingly*

**Changes during revision:** *Line 709-710*

240: Again, it would be worth discussing (briefly) how the shrub dominance of GPP at Mer Bleue might cause it to respond differently to other peatlands, see discussion in https://doi.org/10.1016/j.scitotenv.2018.11.151

L339: Other sites see very large changes, see https://doi.org/10.1111/jvs.12602

L341: suggest also the following as a site where there has been major changes in GPP and plant cover following WT lowering https://doi.org/10.1016/j.scitotenv.2019.134613

**Response**: *We looked those references in detail, The suggested Ratcliffe et al. (2019; 2020) studies, however, have a much longer drainage period (drainage and conversion to pasture in the 1930s and by 1949 conversion to agricultural) than our site at Mer Bleue, yet the before the WTD draw down the water table is always close to the surface in their site. The site was also much nutrient rich than Mer Bleue. We thus decided not to include those studies.*

L384-385: These models are almost totally useless without incorporating feedbacks.

L387: This is a reasonable statement, I agree

L390: I'd say this is not as well established as it might be believed there are odd sites such as pocosin and restiad peatlands where C accumulation can be high even under a very low water table again see: https://doi.org/10.1016/j.scitotenv.2018.11.151

**Response**: *We have rephrased to make it clear in our revision.*

**Changes during revision:** *Line 413-414*

*Lafleur, P., Moore, T., Roulet, N., and Frolking, S, 2005, Ecosystem respiration in a cool temperate bog depends on peat temperature but not water table, Ecosystems 8, 619-629.*

*Peltola, O., Aslan, T., Ibrom, A., Nemitz, E., Rannik, Ü., and Mammarella, I.: The high-frequency response correction of eddy covariance fluxes – Part 1: An experimental approach and its interdependence with the time-lag estimation, Atmos. Meas. Tech., 14, 5071–5088, https://doi.org/10.5194/amt-14-5071-2021, 2021*

**Reply to referee comment (#2) on HESS-2021-585**

*We would like to thank Anonymous Referee #2 for reviewing our paper and for the constructive comments. In the following I listed the referee comments in normal and our reply and list of changes during revision are in italic text.*

I apologize with the Authors for the time it took to provide my review. I enjoyed serving on this study and I do think it is of definite interest. My main concerns are related to the quantification of the uncertainties associated with model parameters and the way they can propagate to model results. I do think that a detailed discussion on this element can strengthen the quality of the results obtained. In the absence of such a quantification, I do think the quality of the model results is at best undetermined. It is with this spirit, and to provide the Authors with ample time to design their revisions, that I am recommending a set of revisions that can range from moderate to major (depending on the way the Authors decide to address these).

While the Authors state that the model they rely upon can lead to a reasonable match with available observations, they offer only limited insights about uncertainties associated with estimated model parameters. Additionally, I do think that the type of sensitivity analysis performed by the Authors does not provide too much quantitative insights about the relative importance of model parameters and I am not entirely sure if the Authors can rely on their results to rank importance of typically uncertain model parameters and the way this impacts model results. I do think at least some discussion on these elements should be included so that the readers can have a full picture at their disposal.

As an additional point of example, it is not clear how the uncertainty associated with parameters governing partially saturated flows (which can be marked while these can be spatially heterogeneous) can impact the quality of the results obtained by the Authors.

The lack (or partial lack) of a rigorous uncertainty quantification in this sense is an element that in my view hampers the way we can quantify the quality of model forecasts. The emphasis that is given to the model performance could be retuned in light of this element, which is critical, in my view.

The Authors find and discuss that NEE displays a linear relation with the water level at the site analyzed. Can they provide some physical meaning to such a linearity? Can in their view this result be transferred to other sites? Perhaps this discussion is already included in the study and I missed these details. In this case, I do apologize with the Authors.

Can the Authors include some details about measurement uncertainties and their view about these can impact model parameter estimation through model calibration?

**Response**: *We will add a section in the supplementary to quantify the parameter uncertainties for modeling continental bog systems and discuss how it would influence our findings on the water table and NEE regulation. However, given our current results are evaluated with 21 years of energy, hydrology, and C fluxes, and the fact that most of the parameters are obtained from many earlier investigations for this site (references for the parameters listed in Table S1 in Supplementary Information), we anticipate that the influence of parameter uncertainties on our key findings will be minor.*

*CoupModel has been applied previously to several boreal peatlands thus parameter uncertainties associated with hydrology and C fluxes were already quantified, e.g., Metzger et al. (2015, 2016), He et al. (2016) and Kasimir et al. (2021). These studies provide prior information for calibration and uncertainty quantification. However, these earlier studies were conducted on fen peat or drained bogs. We expect the following parameters at Mer Bleue would differ from the peatlands evaluated before:*

*Saturated hydraulic conductivity (ksat); maximum stomata resistance of shrubs; the maximum rooting depth of the shrubs; phenology parameters that regulate the start of photosynthesis; the parameter regulating the decrease of photosynthesis when water table depth drops; the*

*decomposition rate of labile and resistant C, (kl and kh); temperature response (i.e., Q10); and soil moisture response for decomposition rate.*

*These are known key parameters that cover the hydrological, photosynthesis, and respiration processes. We will add a calibration using the GLUE approach (He et al. 2016, He et al. 2021) to rank the parameter sensitivity in controlling the water table and NEE.*

*The peat water retention characteristics at Mer Bleue and other similar bog sites have been summarized by Letts et al. (2000). More recently, our peat water retention curves are also shown to be well constrained by a meta-analysis by Liu and Lennartz (2019) who reviewed the existing peat literature. Thus, we have left those coefficients out of the uncertainty analysis.*

*The unique long-term data at Mer Bleue revealed that NEE had a close to a linear relationship with the growing season water level at the annual scale. We have discussed this and expect this to be a general case for other bog sites, see our Discussion (Lines 370-380). This relationship reveals the self-regulation of the bogs that show strong coupling by water storage and growth- also see our Introduction (Lines 36-44).*

*For measurement uncertainty, we will also add a section in SI to describe the eddy covariance processing methodology and the uncertainty related to gap filling. The uncertainty associated with water table depth is documented in the Wilson thesis (2012). We will use those uncertainties to determine the thresholds of accepted model simulations when conducting our GLUE calibration.*

**Changes during revision:** *We have added a comprehensive uncertainty analysis using GLUE methodology in the supplementary section F: Line 173-236. We also have added the supplementary section E to describe the method for flux processing and the potential uncertainty associated with that processing: Line 151-169.*

*The texts were further added into the main paper, Line 153-155, Line 194-197*

*Results of GLUE calibration were added in Line 297-303*

*Metzger, C., Jansson, P.-E., Lohila, A., Aurela, M., Eickenscheidt, T., Belelli-Marchesini, L., Dinsmore, K. J., Drewer, J., van Huissteden, J., and Drösler, M.: CO₂ fluxes and ecosystem dynamics at five European treeless peatlands – merging data and process oriented modeling, Biogeosciences, 12, 125–146, https://doi.org/10.5194/bg-12-125-2015, 2015.*

*Metzger, C., Nilsson, M. B., Peichl, M., and Jansson, P.-E.: Parameter interactions and sensitivity analysis for modelling carbon heat and water fluxes in a natural peatland, using CoupModel v5, Geosci. Model Dev., 9, 4313–4338, https://doi.org/10.5194/gmd-9-4313-2016, 2016.*

*Hongxing He, Per-Erik Jansson, Magnus Svensson, Astrid Meyer, Leif Klemedtsson, Åsa Kasimir, Factors controlling Nitrous Oxide emission from a spruce forest ecosystem on drained*

*organic soil, derived using the CoupModel, Ecological Modelling, 321, 46-63, https://doi.org/10.1016/j.ecolmodel.2015.10.030, 2016,*

*Kasimir Å, He H, Jansson P-E, Lohila A and Minkkinen K (2021) Mosses are Important for Soil Carbon Sequestration in Forested Peatlands. Front. Environ. Sci. 9:680430. doi: 10.3389/fenvs.2021.680430, 2021.*

*Letts, G. M., Roulet, N. T., and Comer, N. T.: Parametrization of peatland hydraulic properties for the Canadian land surface scheme, Atmosphere Ocean, 38, 141-160, 2000.*

*Liu H., Lennartz, B. Hydraulic properties of peat soils along a bulk density gradient- a meta study, Hydrological Processes, 33 (1) 101-114, 2019.*

*Wilson, P.: The relationship among micro-topographical variation, water table depth and biogeochemistry in an ombrotrophic bog, Master of Science thesis, McGill University, 2012.*

Hongxing He

On behalf of all authors